# Antivenom Administration After Rattlesnake Envenoming in Arizona Does Not Directly Diminish Pain

**DOI:** 10.3390/toxins16120521

**Published:** 2024-12-02

**Authors:** Vance G. Nielsen, Darien L. Stratton, Tyler M. Hoelscher, Hannah L. Nakamura, Matthew M. Cornelison, William F. Rushton, Geoffrey T. Smelski

**Affiliations:** 1Department of Anesthesiology, The University of Arizona College of Medicine, Tucson, AZ 85724, USA; 2Division of Medical Toxicology, Department of Emergency Medicine, University of Pittsburgh School of Medicine, Pittsburgh, PA 15213, USA; strattondl@upmc.edu; 3Poison Control Center, The University of Arizona College of Pharmacy, Tucson, AZ 85724, USA; thoelscher@arizona.edu (T.M.H.); hnakamur@arizona.edu (H.L.N.);; 4Office of Medical Toxicology, Department of Emergency Medicine, UAB Medicine, The University of Alabama at Birmingham, Birmingham, AL 35249, USA; mcornelison@uabmc.edu (M.M.C.); wrushton@uabmc.edu (W.F.R.)

**Keywords:** pain, rattlesnake, envenomation, antivenom, analgesia

## Abstract

The onset, progression, and severity of pain following rattlesnake envenomation are highly variable between patients. Pain can be severe and persistent, seemingly refractory to opioid analgesics. The ability of antivenom to directly relieve pain has not been well studied. We reviewed poison center charts of rattlesnake envenomations between 1 January 2018, and 31 December 2022. Demographic data as well as details of antivenom usage and pain severity were collected. Patients were coded in one of three categories: without pain (Pain 0), well controlled pain (Pain 1), and opioid refractory pain (Pain 2). A total of 289 patients met the inclusion criteria, with 140 receiving Anavip antivenom and 149 receiving Crofab. Patient characteristics were different between both cohorts. There were no significant differences in the number of Anavip vials used between the Pain 1 and Pain 2 groups. However, patients in the Crofab Pain 2 group received more antivenom compared to Pain 1. Importantly, Pain 3 patients were treated with the highest amount of antivenom in both the Crofab and Anavip cohorts. Despite the higher doses of antivenom used, these patients also experienced the greatest pain. These data suggest that antivenom alone may have minimal analgesic benefits.

## 1. Introduction

The onset, progression, and severity of pain occurring after a venomous snakebite is dependent on the venom proteome and preformed compounds as recently reviewed [1]. Across the globe, nearly all snake envenomations are associated with immediate pain, independent of the mechanical injury from the fangs [2]. Instead, pain is caused by various toxins found within venom such as biogenic amines and enzymes including phospholipase A_2_ and metalloproteinases [3,4,5,6,7,8,9,10,11,12,13,14,15,16,17,18,19]. As tissue injury at the bite site evolves, multiple byproducts of enzymatic activity activate several pain-modulating receptors on peripheral nerves, with amplification of pain intensity occurring [1]. While relatively painless bites by snakes with neurotoxic venom may cause remote pain syndromes (e.g., abdominal pain, headache) [1], North American pit vipers generally inflict the greatest pain at the bite site and along the border of ongoing tissue injury that extends from the bite [1,2,20,21]. Figure 1 displays the various compounds and enzymes found in rattlesnake venom and the receptors stimulated on nociceptive nerves as reviewed [1].

The relationship between antivenom and pain relief has not been well characterized. In the case of mild to moderate severity copperhead bites, a small placebo-controlled trial found patients who received antivenom, also received fewer opioid doses compared to the placebo group [22]. To date, no study has reported on the effect antivenom has on rattlesnake envenomation-induced pain, although it is of great clinical interest. Opioid administration is not benign and the potential for opioid dependency during recovery from envenomation exists. If antivenom does indeed have analgesic effects, then there may be an opportunity to limit opioid use following a rattlesnake bite. However, on a cellular level, antibody fragments used as antivenom have not been demonstrated to directly interact with pain pathways and antivenom-associated analgesia would not be expected to occur in poorly perfused, edematous bite sites secondary to damage that has already occurred. Prior reports of antivenom-associated analgesia are sparse, with literature instead emphasizing the benefits of other multimodal pain control [23,24,25,26].

To fill this critical gap in knowledge, the goal of the present study was to elucidate, retrospectively, any relationship between antivenom administration and the severity of pain/analgesic intervention following envenomation by Arizonan rattlesnakes.

## 2. Results

### 2.1. Patient Population Analyzed After Rattlesnake Bite

This was a retrospective, observational investigation that was approved by our institutional review board. As detailed in Methods, patient records were assessed for all cases coded as a rattlesnake bite receiving antivenom by one author that originated from the Arizona Poison and Drug Information Center from 1 January 2018 to 31 December 2022. The salient details of patient characteristics and clinical management were placed into a database after patient de-identification was performed.

After identifying 587 potential bite victims, exclusion criteria were applied as detailed in Methods, with 289 patient records found to meet the criteria. Thereafter, the records were divided by antivenom administered, as the loading dose of Anavip antivenom is 10 vials and Crofab antivenom is 4–6 vials to treat tissue damage. This resulted in two databases with 140 patients administered Anavip and 149 patients administered Crofab as depicted in Figure 2.

The salient characteristics of both populations are displayed in Table 1. There were no significant differences in age, gender, or incidence of bite by anatomical location between the two populations.

### 2.2. Quantification of Antivenom Administration (Vials) per Patient Stratified by Antivenom Type and Degree of Pain

The initial number of loading doses for tissue damage, the initial number of vials administered for tissue damage, and the total number of vials during hospitalization are depicted in Figure 3 (Anavip) and Figure 4 (Crofab). The administration of each antivenom is further separated by the degree of pain experienced by the patient. Patients were categorized as those without pain (Pain 0), well controlled pain (Pain 1), and opioid refractory pain (Pain 2). Pain severity and opioid usage were assessed during the first two days of treatment.

As displayed in Figure 3, there were no significant differences in antivenom administration between Pain 0 and Pain 1 groups; however, Pain 2 patients had significantly greater administration of Anavip vials than the other two groups. In the case of Crofab administration, these data are presented in Figure 4.

The pattern of Crofab antivenom administration and associated pain score was like that of Anavip, except for the total vials administered. There was a significant increase in vials of Crofab administered during hospitalization with every increase in pain score.

## 3. Discussion

The present investigation achieved its stated goal. These data indirectly support the concept that the primary driver of pain following rattlesnake bite in Arizona is tissue damage by PLA_2_ and metalloproteinases [1,11,12,13,14,15,16,17,18,19], as antivenom is administered to minimize further tissue damage, not pain. Put another way, patients with the greatest pain were administered the greatest amount of antivenom to treat their injury. This pattern persisted following the administration of two different antivenoms, strengthening the contention that there is no analgesic benefit to the antivenoms as previously posited [22]. However, to the extent that antivenom administration prevented ongoing tissue damage, it likely diminished the potential progression of pain severity. In summary, the severity of pain and need for opioid therapy is determined by the extent of tissue damage following envenomation and not determined directly by antivenom administration.

Prior literature on the analgesic effects of rattlesnake antivenom is severely lacking. However, a variety of pain syndromes inflicted by other species of venomous snake are attenuated by antivenom administration as recently reviewed [1]. Examples include nonspecific chest pain, abdominal pain, and generalized body pain that are inflicted by Elapidae species known to be neurotoxic [1]. However, antivenom is not a uniquely snake-specific treatment for envenomation-associated pain. Pain associated with envenomation by the Latrodectus species makes an interesting comparison to Elapidae. As the clinical presentation following Latrodectus envenomation involves more pain than evidence of tissue destruction [27], research exploring the effectiveness of the Latrodectus antivenoms revolves around the analgesic effectiveness of the antivenom. Multiple prospective randomized controlled trials have been conducted in the United States and Australia on Latrodectus species antivenom-related analgesia [28,29] but unfortunately have yielded poor conclusions about the antivenom’s clinical benefit compared to the clinical benefit of Elapidae antivenoms as reviewed [1]. Further investigation is needed to determine if antivenoms function by removing the neurotoxins involved from the circulation or perhaps could function as weak analgesics by yet-to-be-defined mechanisms.

This investigation was limited, as all retrospective studies are, by the ability to prospectively control the numbers of patients recruited to balance the three pain scores and control other potentially important factors. However, the numbers observed in the three pain scores were preserved proportionately in both antivenom groups—thus, the incidence of Pain 0, Pain 1, and Pain 2 may very well reflect reality prospectively. Considered as a whole, despite these limitations, it is reasonable to find the relationship between pain score and antivenom administration plausible from the data presented.

Since antivenom administration does not seem to be directly useful for analgesia, and a significant number of patients still suffer severe pain despite considerable medical intervention, these data serve as a “call to arms” to more proactively design a multimodal approach to analgesia in such clinical circumstances. Such a strategy could include appropriate analgesic medication, environmental modification, limb elevation, and regional anesthesia, when appropriate. Thus, our data serve as a rationale to continue to refine methods to improve analgesia after rattlesnake bite not just in Arizona but wherever rattlesnakes or other vipers with similar venom proteomes may inflict envenomation-mediated pain.

In summary, our retrospective investigation demonstrated that pain caused by rattlesnake bites in Arizona is not directly affected by the quantity of antivenom administered for tissue damage. The management of opioid refractory pain in this setting requires strategies in addition to antivenom administration. These strategies will likely be multimodal, including regional analgesia [23,24,25,26] or administration of medications such as ketamine [30]. Further investigations are needed to determine the optimal approach for crotalid envenomation-related pain.

## 4. Materials and Methods

### 4.1. Rattlesnake Bite Database

The Arizona Poison and Drug Information Center serves the state of Arizona, excluding Maricopa County. The Center’s standard practice for managing rattlesnake bites includes recommending antivenom for all patients with signs of envenomation. The Center’s records for all cases coded as a rattlesnake bite receiving antivenom were searched between 1 January 2018 and 31 December 2022. Records were not included when poison center involvement was incomplete, such as when patient care was transferred to another regional poison center, the patient left the hospital against medical advice, or when the poison center was not contacted until after the original hospital visit. Cases were excluded from analysis when they lacked sufficient details for the study outcomes. A standardized study template consisting of basic demographic and clinical data was utilized.

Patient records were manually reviewed by a single, unblinded data abstractor (G.T.S.) to assess for the presence of pain and opioid administration, and data were abstracted into a standardized template. Ambiguous cases were coded negative (e.g., if it was uncertain if the pain was not refractory to opioid administration, then the case was coded as Pain 1, not Pain 2). Pain scoring in the first two days of hospitalization was as follows: Pain 0 = without pain; Pain 1 = pain well controlled by opioids; Pain 2 = ongoing, severe pain despite repeated doses of opioids.

The inclusion/exclusion criteria were designed to provide the ability to clearly assess the degree of pain in relation to a limited set of envenomation conditions. Thus, the exclusion criteria included patients <18 years of age, patients with chronic pain or with a previous injury in the bitten extremity, patients that received antivenom >4 h after being bitten, patients administered both antivenoms, patients with dementia, patients with diabetes (potential preexisting neuropathy), and patients not bitten on an extremity (e.g., abdominal wall, face). Data were further stratified by antivenom administered, given the differences in the number of recommended vials per loading dose between CroFab^®^ (4–6 vials), and Anavip^®^ (10 vials).

### 4.2. Rattlesnake Bite Treatment Based on Tissue Damage

In consultation with the Arizona Poison and Drug Information Center, health care providers administered antivenom to treat tissue damage. Diagnosis of envenomation is based on the history provided in conjunction with the development of clinical symptoms, which include systemic toxicity, coagulopathy, and local tissue injury. Diagnosis is almost always made based on the presence of a clear local tissue injury. Once any pain, edema, erythema, and/or ecchymosis are present beyond what would be expected from a needle puncture, it is considered sufficient to diagnose envenomation and antivenom is ordered. Antivenom is recommended for all envenomations, the Arizona Poison and Drug Information Center does not recommend withholding antivenom for potentially minor envenomations because of the preventative nature by which antivenom functions.

The antivenoms administered were either crotalidae polyvalent immune fab-ovine (Fab) [CroFab^®^, BTG International Inc., Conshohocken, PA, USA] or crotalidae immune F(ab’)_2_ [Anavip^®^, Rare Disease Therapeutics Inc., Franklin, TN, USA]. After administration of a loading dose, ongoing administration was based on whether the signs of ongoing tissue damage stopped (e.g., if edema continued to evolve >1 inch/h, pain was worsening, or new blisters were developing, additional antivenom was administered).

### 4.3. Statistical Analyses and Graphics

Data are presented as sums or mean ± SD. A commercially available statistical program was used for unpaired, two-tailed Student’s *t*-test, chi-square, and one-way analyses of variance (ANOVA) as appropriate to the dataset, followed by Holm–Sidak post hoc analyses (SigmaStat 3.1; Systat Software, Inc., San Jose, CA, USA). Graphics were generated with commercially available programs; CorelDRAW 2024, Alludo, Ottawa, ON, Canada; Origen 2024, OrigenLab Corporation, Northampton, MA, USA; and BioRender, Toronto, ON, Canada). *p* < 0.05 was considered significant.

## Figures and Tables

**Figure 1 toxins-16-00521-f001:**
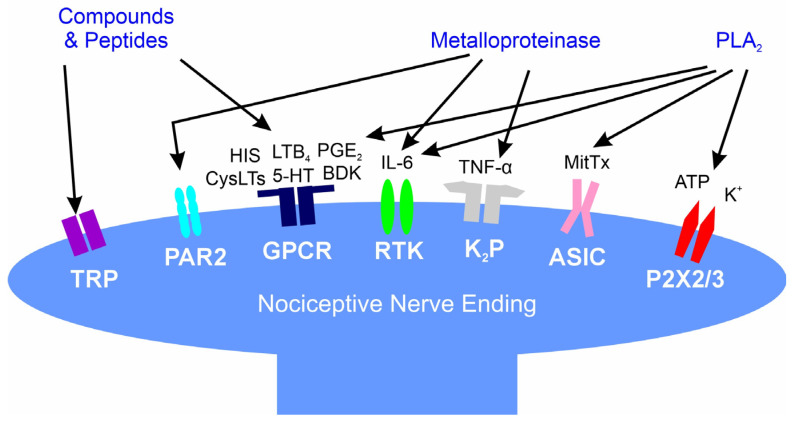
Interactions of snake venom compounds and proteins with nociceptive nerve endings and other key systems that result in pain. As explained in detail [1], the indicated compounds and proteins activate receptors either directly or via products of enzymatic catalysis. ASIC—acid-sensing ion channel; ATP—adenosine triphosphate; BaP1, Batroxase, BpirMP—examples of metalloproteinases; BatroxPLA_2_, Lemnitoxin—examples of PLA_2_; BDK—bradykinin; Ch—choline; CysLTs, LTB_4_—examples of leukotrienes; GPCR—G-protein coupled receptor; HIS—histamine; IL-6—interleukin 6; K^+^—potassium; K_2_P—two-pore potassium channel; MitTx—a low activity PLA_2_ molecule bound to a with Kunitz-like protein that directly activates ASIC; P2X2/3—purinoceptors 2X2 and 2X3; PAR2—protease-activated receptor 2; PGE_2_—prostaglandin E_2_; RTK—receptor tyrosine kinase; and, TNF-α—tumor necrosis factor-α; TRP—transient receptor potential channel.

**Figure 2 toxins-16-00521-f002:**
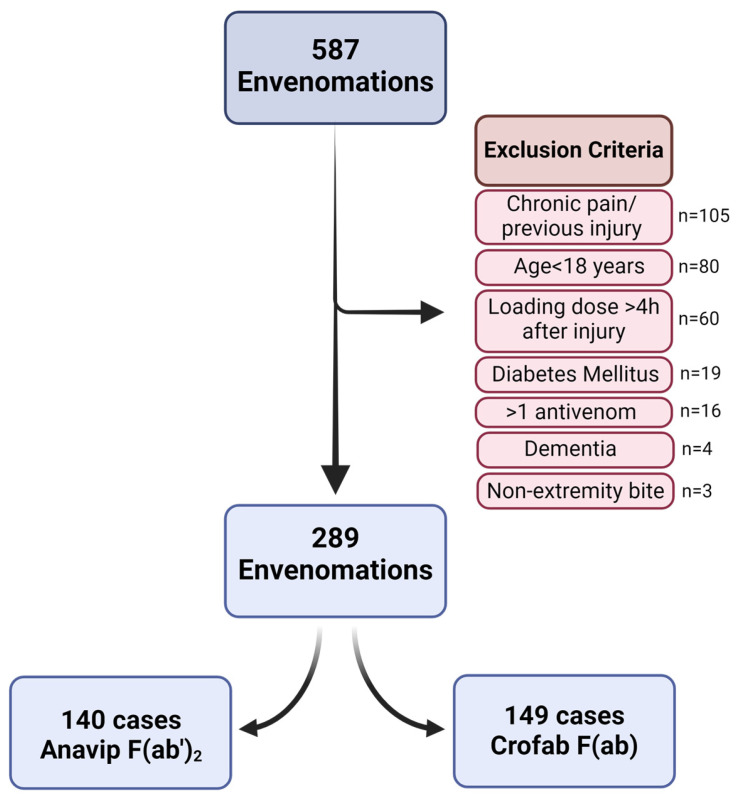
Patient selection flow chart.

**Figure 3 toxins-16-00521-f003:**
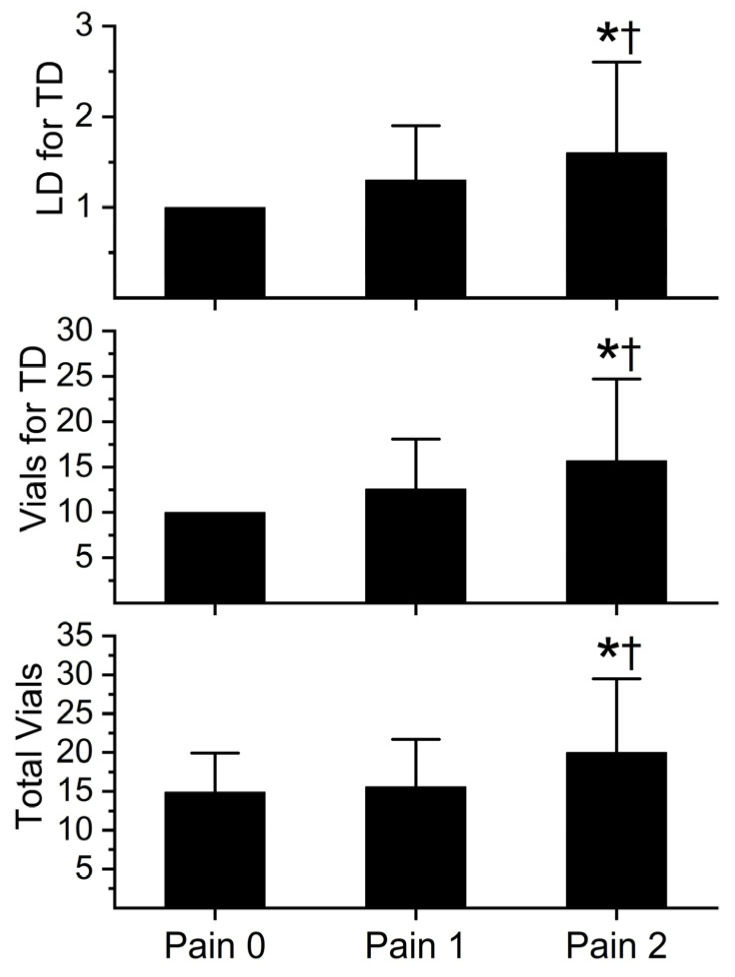
Anavip vial administration and pain score. LD = loading dose; TD = tissue damage. Data presented as mean ± SD. Data were analyzed with one-way analysis of variance (ANOVA) with Holm–Sidak post hoc test. * *p* < 0.05 vs. Pain 0; † *p* < 0.05 vs. Pain 1. Pain 0, n = 9; Pain 1, n = 87; Pain 2, n = 44.

**Figure 4 toxins-16-00521-f004:**
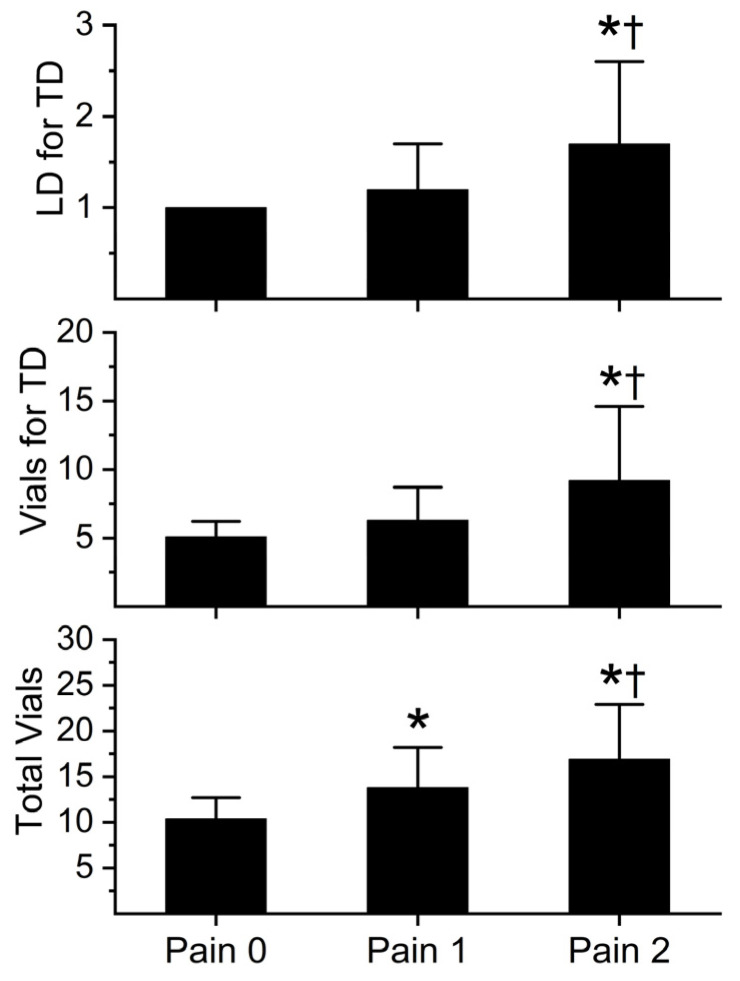
Crofab vial administration and pain score. LD = loading dose; TD = tissue damage. Data presented as mean ± SD. Data were analyzed with one-way analysis of variance (ANOVA) with Holm–Sidak post hoc test. * *p* < 0.05 vs. Pain 0; † *p* < 0.05 vs. Pain 1. Pain 0, n = 11; Pain 1, n = 90; Pain 2, n = 48.

**Table 1 toxins-16-00521-t001:** Characteristics of rattlesnake bite victims stratified by antivenom administered.

Characteristic	Anavip	Crofab	*p* Value
Patient Number	140	149	-
Age (years)	53.8 ± 17.6	52.9 ± 17.2	0.662
Male:Female	77:63	97:52	0.080
Finger Bites	36	53	0.441
Hand/Wrist Bites	17	18	0.987
Arm Bites	1	4	0.199
Toe Bites	17	15	0.574
Foot/Ankle Bites	48	47	0.620
Calf/Thigh Bites	21	12	0.064

Values are depicted as total number or mean ± SD. Age was analyzed with two-sided, unpaired Student’s *t*-test and all other characteristics were analyzed with chi-square.

## Data Availability

The original contributions presented in the study are included in the article and further inquiries can be directed to the corresponding author.

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
