# Peer review of "Antivenom Administration After Rattlesnake Envenoming in Arizona Does Not Directly Diminish Pain"

_toxins, 2024, doi:10.3390/toxins16120521_

Round 1
Reviewer 1 Report
Comments and Suggestions for Authors
The study is investigating an unrevealed potential of antivenom in pain relief during envenomation. The study is relevant since snake venoms are know to be rich in peptides and molecules acting on the nervous system and more precisely sensory system. In addition, toxicity triggered by snake venoms promote pain. The study is well designed however, no correlation was found between antivenom administration and pain relief.
Authors are suggested to use more references related to the topic as the majority of referencing is linked to ref.1.
The introduction requires more details about the literature related to your study.
Adding a figure explaining pain sensing upon envenomation may facilitate the understanding.
The discussion should be improved as a reader that is not familiar with the topic may feel lost. Adding references conducting similar research on other snakes / antivenom is recommended.
Author Response
Reviewer #1
“Comments and Suggestions for Authors
”
“The study is investigating an unrevealed potential of antivenom in pain relief during envenomation. The study is relevant since snake venoms are know to be rich in peptides and molecules acting on the nervous system and more precisely sensory system. In addition, toxicity triggered by snake venoms promote pain. The study is well designed however, no correlation was found between antivenom administration and pain relief.”
We appreciate the reviewer’s comments. We agree that antivenom did not correlate with pain relief; rather, pain relief was the worst in patients administered the most antivenom. Thus, antivenom is not a direct analgesic but instead diminishes pain to the degree that tissue damage is treated.
“Authors are suggested to use more references related to the topic as the majority of referencing is linked to ref.1.”
We used ref. 1. As it is encyclopedic and a manuscript that has been viewed almost 8,000 times. However, we are happy to include several additional references in the hopes of satisfying the reviewer.
“The introduction requires more details about the literature related to your study.”
We have tried to provide what little is known about rattlesnake envenomation and pain – which is not much. We hope that providing additional molecular concepts in figure and reference form will satisfy the reviewer.
“Adding a figure explaining pain sensing upon envenomation may facilitate the understanding.”
We strongly agree with the reviewer and now include a figure that is more customized towards rattlesnake venom as our new figure 1.
“The discussion should be improved as a reader that is not familiar with the topic may feel lost. Adding references conducting similar research on other snakes / antivenom is recommended.”
We wish that we could add additional studies that are similar in nature to ours. However, as we note in the Discussion, there are essentially no other studies that have addressed the issue that is central to our manuscript. We certainly do not want the reader to feel lost, but we are unable to provide references similar to our work. It is our hope that the reviewer is satisfied with the additional references and figure to better acquaint the readership with the mediators of pain involved in rattlesnake envenomation in our state.
Reviewer 2 Report
Comments and Suggestions for Authors
1. The main question addressed by the research:
The manuscript investigates the role of antivenom therapy in pain relief following rattlesnake bites. The retrospective analysis of cases managed by Arizona Poison and Drug Information Center investigates the pain levels of patients treated with different doses of two antivenoms.
2. The new content of the subject area compared with other published material:
Regarding the scientific literature, one would not expect direct analgetic effects of antivenom therapy on the site of bite other than that caused by reduction in tissue damage. The findings suggest no connection between the antivenom therapy and pain relief. However the question is clinically relevant, the findings are not surprising in the light of rattlesnake venom composition.
3. The authors should consider the specific improvements regarding the methodology and further controls:
Despite the sufficient number of data, the filtering of sample did not manage to discriminate between different pain management strategies. Also, no systemic effects were discussed regarding multi-organ involvement in pain induction. It seems, that only bitewound-associated tissue damage and localized pain was evaluated.
4. General discussion about the patient's pain is somewhat confusing without clarification of localisations.
5. In summary: although the investigation has some clinical relevance, without further clarification it has a very limited potential in development of pain management strategies. I would recommend to convert the manuscript in the form of a Short Communication.
Author Response
Reviewer #2
“Comments and Suggestions for Authors
”
“1. The main question addressed by the research:
The manuscript investigates the role of antivenom therapy in pain relief following rattlesnake bites. The retrospective analysis of cases managed by Arizona Poison and Drug Information Center investigates the pain levels of patients treated with different doses of two antivenoms.”
We appreciate the clarity expressed by the reviewer concerning the focus of our investigation. We would simply note that the two antivenoms are administered in different doses as per the manufacturer’s recommendation to a similar endpoint of tissue damage control.
“2. The new content of the subject area compared with other published material:”
“Regarding the scientific literature, one would not expect direct analgetic effects of antivenom therapy on the site of bite other than that caused by reduction in tissue damage. The findings suggest no connection between the antivenom therapy and pain relief. However the question is clinically relevant, the findings are not surprising in the light of rattlesnake venom composition.”
We appreciate the concern of the reviewer that based on biochemistry, one would not necessarily expect a direct analgesic effect of the antivenoms tested. However, one reference (now #22 after additional references included as per the request of the other reviewer) that we cited in our Introduction did lean towards an analgesic effect of these antivenoms in the presence of rattlesnake envenomation. In summary, our data substantiates that the antivenoms administered are not directly analgesic as suggested [22], but rather supports that the degree of attenuation of tissue damage by the antivenoms administered is the mechanism responsible for whatever pain relief is offered.
“3. The authors should consider the specific improvements regarding the methodology and further controls:”
“Despite the sufficient number of data, the filtering of sample did not manage to discriminate between different pain management strategies. Also, no systemic effects were discussed regarding multi-organ involvement in pain induction. It seems, that only bitewound-associated tissue damage and localized pain was evaluated.”
We appreciate the concerns raised by the reviewer. The standard of care for pain relief at this time in our state is the administration of opioids concurrent with antivenom administration and other supportive measures. The particular opioid administered, frequency, dose, etc. is managed by the emergency room physicians overseeing the care of these rattlesnake bite victims. The strategy is to provide the patient with the best analgesia with this approach that is possible. Further, as we mention in our Introduction, the pain associated with rattlesnake envenomation is associated with the bite site and the border of ongoing tissue injury that extends proximally from the bite [1,2,20,21]. Put another way, rattlesnake envenomation is not associated with pain syndromes distant from the bite site, unlike as in the case of venom derived from elapids [1].
“4. General discussion about the patient's pain is somewhat confusing without clarification of localisations.”
Given the nature of pain following rattlesnake envenomation as detailed in point #3 previously, we tried to offer the reader as granular a data set as we could as presented in our table 1. The pain was emanating from the indicated bite sites as indicated anatomically and by antivenom, presented in table 1. We are unsure as to how better address the comment of the reviewer and hope that our response addresses the concern raised.
“5. In summary: although the investigation has some clinical relevance, without further clarification it has a very limited potential in development of pain management strategies. I would recommend to convert the manuscript in the form of a Short Communication.”
It is our hope that our responses to the reviewer have clarified any ambiguity that existed concerning the data and our interpretation of it. As for the development of pain strategies, this issue is beyond the scope of the present work. We did mention the use of multimodal analgesic strategies, including regional nerve block, in our Introduction [23-26]. Now that we can clearly note that antivenom only provides analgesia to the extent that tissue injury is treated, the pursuit of more effective forms of multimodal analgesia following envenomation by Arizona rattlesnakes should occur in future studies. With the addition of new text, figure, and citations, we believe that this work should remain an article.
Round 2
Reviewer 2 Report
Comments and Suggestions for Authors
The clarifications and the added figure made the manuscript somewhat more relevant. The new references are useful.